# Prevalence of unrecognized depression and associated factors among medical outpatient department attendees; a cross sectional study

**Alemu Lemma** [1]*, **Haregewoyin Mulat**[1], **Kabtamu Nigussie**[2], **Wondale Getinet**[1]

**1** Department of Psychiatry College of Medicine and Health Science, University of Gondar, Gondar, Ethiopia,
**2** Department of Psychiatry, School of Nursing and Midwifery, College of Health and Medical Sciences, Haramaya University, Harar, Ethiopia

* emailmeonalex@gmail.com

## Abstract

### Objectives

To determine the prevalence and associated factors of unrecognized depression among patients who visit non-psychiatric outpatient departments in the University of Gondar specialized teaching hospital. North West Ethiopia.

### Methods

An institution-based cross-sectional study was conducted among Medical outpatient departments in the University of Gondar specialized referral hospital from March to April 2019. We collected data through face-to-face interviews. We recruited 314 participants for face-to-face interviews using the systematic random sampling technique. The patient health questionnaire (PHQ-9) was used to measure depression. Coded variables were entered into Epi Info version 7 and exported to SPSS version 20 for analysis. Descriptive statistics and multivariable logistic regression analysis were used. Adjusted odds ratios (AOR) with a 95% confidence interval were used to calculate significance.

### Results

A total of 314 participants were interviewed with a response rate of 100%. The prevalence of depression was 15.9% with (95% CI (12.1–20.1)). In the multivariate logistic regression revealed that, able to read and write (AOR = 0.24, 95% CI (0.67–0.84)), secondary education (AOR = 0.34, 95% CI (0.12–0.91)), education in college and university level (AOR = 0.32, 95% CI (0.13–0.78)), poor social support (AOR = 7.78, 95% CI (2.74–22.09)), current cigarette smokers(AOR = 12.65, 95% CI (1.79–89.14)) were associated with depression.

### Conclusion

The prevalence of depression among outpatient attendees was high. We recommend an early depression screening be carried out by health professionals.

**Data Availability Statement:** All relevant data are available in the Supporting information files and in the Qualitative Data Repository: https://data.qdr.

syr.edu/dataset.xhtml?persistentId=doi:10.5064/
F61LTHZQ.

**Funding:** The author(s) received no specific funding for this work.

**Competing interests:** The authors have declared that no competing interests exist.

**Abbreviations:** AOR, Adjusted Odds Ratio; COR, Crude Odds Ratio; ICCMH, Integrated Clinical and Community Mental Health; LAMIC, Low and Middle-Income Countries; MDD, Major Depressive Disorder; PHQ, patient health questionnaire; SPSS, Statistical Package for Social Science; WHO, World Health Organization.

## Introduction

The global report shows that near to 500 million people worldwide have mental illness and 25% meet the criteria of mental illness at some point in their life [1, 2], depression alone affects 350million people worldwide [3], and is the second leading cause of disability, depressed individuals have 20 times greater suicide than the general population [4, 5].

Depression is among the most common mental disorders characterized by sadness, loss of interest or pleasure, guilt feeling or low self-esteem, disturbed sleep or appetite, decreased energy, and poor concentration [6]. Depression is common in life and can be in the form of blues or sadness, mourning, or in the form of hyperactivity and manic behavior [7]. A one time, one-year and lifetime prevalence of depression among world population were reported to be 12.9%, 7.2% and 10.8% respectively [8]. In Ethiopia prevalence of depression was reported to be 9%, and is the 7th leading cause of disease burden [6].

World health organization figure on mental health shows that undiagnosed depression places high socioeconomic burden on individuals, families and community in terms of quality of life, increase medical morbidity and mortality, leads disability, reduce occupational performance [9].

Comorbidity of depression with chronic medical conditions like diabetes, hypertension, asthma, sickle cell disease, cardiac diseases, chronic respiratory diseases and rheumatoid arthritis is very common [10]. Studies reported as depression comorbidity with medical case may result in prolonged hospital admission, increase physical symptoms, reduction in adherence to medical treatment and increased medical costs [7, 9].

Another Study conducted on clients visiting medical outpatients reported as 5.4% had major depressive disorder [11], those attending geriatric outpatients 53.2% had depression [12], adult primary care 10.7% had depression [13], 23.8% India [14], 60.5% in Jamaica [15], 49.8% in Nigeria [16], 30.3% in Malawi [17], 38% Hawassa Ethiopia [18], 32.2% in Ambo Ethiopia University [15].

Different risk factors results for depression such as personal, social, psychological, environmental, chronic medical illness, family history of mental health problems, exposure to violence and crime [19–27]. However, there are limited data in the country particularly unrecognized depression among medical patients where psychiatric services is limited and or not available is under studied. Therefore, this study aimed to assess the magnitude and associated factors of unrecognized depression among medical outpatient attendees. This would help for future integrated intervention and it would be an input of information for policymakers to think of intervention strategies.

## Materials and methods

### Study design, periods and study area

An institution based cross sectional study design was employed from 22nd March to 30th April 2019. The survey was conducted at the University of Gondar comprehensive specialized hospital. The University of Gondar hospital is in the Northwest part of Ethiopia near to Sudan border. It is a tertiary level referral hospital, which acts as the referral center for over ten district hospitals in the area. The hospital has seven adult outpatient clinics and 600 inpatient beds, and 850 health professionals to provide health service to the community. Majority of professionals are nurses (n = 500). This hospital gives health referral services over 5 million inhabitants in the Northwest region of Ethiopia.

### Sample size and sampling procedure

The participants of this study were patients receiving outpatient care at University Gondar compressive specialized hospital, Gondar, Ethiopia. We use a single population proportion

formula, $n = Z^2p(1 - p) / d^2$ with a 5% margin of error, 95% confidence level and with the assumption prevalence(P) of depression 24.5% [7] used to calculate sample size yielded 314 (adding 10% non-response rate). The average number of patients was calculated with previous monthly visit in mind participants for interviews. A systematic random sampling technique was used to select the study participants for interview. A total of 58,300 and 4,858 clients attend the medical OPD annually and monthly, respectively. The sampling fraction (K) was obtained by dividing monthly average number of patients attending medical outpatient department for the sample size, which is 15. The first individual was selected using a lottery method, and the rest were selected at a regular interval using systematic random sampling method.

## Inclusion and exclusion criteria

All patients who attended adult medical OPD at University of Gondar compressive specialized hospital were the source population, and those who were attending adult medical OPD at University of Gondar compressive specialized hospital during the study period and who fulfilled inclusion criteria were considered as study population. Clients who were already diagnosed with depression, unable to communicate during the interview as a result of critical illness were excluded from this study.

## Data collection tools and procedures

Depression among patients visiting at outpatient departments for the last two weeks was assessed by the Amharic version of Patient health questionnaire (PHQ9). A PHQ-9 measurement ranges from zero to three. It has demonstrated acceptable reliability and validated to use in Ethiopia for screening depression [28]. A cut of point of ten and above was used for depression. A PHQ-9 include the DSM V depression criteria along with other leading depression symptoms into a brief self-report scale [28]. Social support was measured by Oslo Social Support scale, it covers different fields of social support by measuring the number of people the respondent feels close to, the interest and concern shown by others, and the ease of obtaining practical help from others [2] the scale ranged from 3–14 and the scores 3–8, 9–11 and 12–14 stood for po7or, moderate and strong social support respectively. Unstandardized semi structured questionnaires used to assess substance use, socio-demographic and clinical factors. Data were collected by face to face interviews using a semi structured questionnaire by three trained psychiatry nurses by the Amharic version of the tool. First, questionnaires were designed in English and translated to Amharic for interview and back translation to English was performed by another expert to ensure its consistency with the original version and check its understandability. Data collectors were trained for one day, about research methods, interviewing skills and ethical aspects of the research.

## Data processing and analysis

All data were collected by using Interviewer administered technique. The completeness and consistency of questionnaires were manually checked. The data were coded and entered into Epi-Info version 7 and exported to SPSS for further analysis. Descriptive and bivariate logistic regression analyses were computed to see frequency distribution and to test the association between independent and dependent variables, respectively. Factors associated with depression were selected during bivariate analyses with a p-value less than or equal to 0.2 for further multivariate analysis in which variables with less than 0.05 p-value at a 95% confidence interval were considered as statistically significant.

### Ethics approval and consent to participate

Ethical clearance was obtained from the University of Gondar Institutional Review Board following the Ethiopian National Research Ethics Review Guideline of the Federal Ministry of higher education and Science. A permission letter was obtained from the Gondar referral hospital. The study was performed in accordance with the declaration of Helsinki. Participants were informed about the aim of the study, procedures of selection, and assurance of confidentiality, their names were not registered to minimize social desirability bias and enhance anonymity. The right to participate, to refuse or discontinue participation at any time they want and the chance to ask any thing about the study was given for the participants. Informed written consent was obtained from all participants.

## Result

### Socio demographic factors of study participants

A total of 314 respondents were participated with a response rate yielding 100%. More than half, 164 (52.2%) were male. The mean age of respondents was 32.75 (SD = 11.7) years. Nearly half 147(46.8%) were married, and 144(45.9%) live with husband /wife, while 137(43.6%) of participant were single. Less than half respondents 120(38.2%) were private workers followed by 74(23.6%) government employee.

Among the participants 122(38.9%) had College/university level of education. The majority of respondents, 254(80.9%) were Orthodox Christians (Table 1).

A 115(36.6%) of respondent had poor social support, whereas 102(32.5%) had moderate and 97(30.8%) had good social support. Small number of respondents 23(7.3%) used khat leaves at least once in their life time and20 (6.4%) used khat leaves in the last three months. About 57(18.2%) ever consumed alcohol in their life time and 54(17.2%) consumed alcohol in the last three months. In addition, 8(2.5%) were ever smoking cigarette in their life time, and 6 (1.9%) were smoking cigarette in the last three months (Table 2).

### Prevalence of unrecognized depression

As per PHQ-9 15.9% [(95%CI (12.1–20.1)] of the medical outpatient attendees were identified to have unrecognized depression.

### Factors associated with depression

To determine the association of independent variables with unrecognized depression, bivariate and multivariate binary logistic regression analyses were done. In the bivariate analyses, factors including, educational status, living condition, substance use and social support were significantly associated with depression at a p-value less than 0.2. These factors were entered into the multivariable logistic regression model to control confounding effects. The result of the multivariate analysis showed that able to read and write, being secondary education, being college and university education, being current cigarette user and poor social support were significantly associated with depression at a p-value less than 0.05. Being able to read and write were 76% times less likely for depression than illiterate (AOR = 0.24, 95%CI (0.67–0.84)). Being at level of secondary education were 66% times less likely to develop depression compared to illiterate (AOR = 0.34, 95%CI (0.12–0.91)). Being in college/university education status were 68% times less prone to depression than illiterate (AOR = 0.32, 95%CI (0.13–0.78)). The odds of developing depression were 7.78 times higher among people with poor social support than strong social support (AOR = 7.78, 95%CI (2.74–22.09)). The odds of developing depression

were 12.65 times higher among respondents in current cigarette smoker than non-smokers (AOR = 12.65, 95% CI (1.79–89.14)) (Table 3).

## Discussion

The prevalence of depression among respondents on this study was 15.9% [95%CI: (12.1–20.1)]. Our finding was consistent with cross sectional studies conducted in Debretabor Ethiopia (17.5%) [3] this might be because of we employed the same tool. But, our finding was higher than the finding of a systematic review noted in Ethiopia which had used Composite International Diagnostic Interview(6.8%) [29]. The possible explanation for higher prevalence of depressive episodes in our study might be due to the methods used we conducted cross

**Table 1. Socio-demographic characteristics of the study participants (n = 314).**

| Variables | Frequency | P (%) |
|---|---|---|
| Sex: - | | |
| Male | 164 | 52.2 |
| Female | 150 | 47.8 |
| Occupation: - | | |
| Farmer | 60 | 19.1 |
| House wife | 41 | 13.1 |
| Governmental worker | 74 | 23.6 |
| Private worker | 120 | 38.2 |
| Student | 19 | 6.1 |
| Marital status: - | | |
| Single | 138 | 43.9 |
| Married | 147 | 46.8 |
| Divorced | 29 | 9.2 |
| Religion: - | | |
| Orthodox | 254 | 80.9 |
| Muslim | 32 | 10.2 |
| Protestant | 25 | 8 |
| Catholic | 3 | 1 |
| Educational status: - | | |
| Unable to read and write | 48 | 15.3 |
| Able to read and write | 37 | 11.8 |
| Primary education | 44 | 14 |
| Secondary education | 63 | 20.1 |
| College/university | 122 | 38.9 |
| Living status: - | | |
| Living alone | 72 | 22.9 |
| With parents | 69 | 22 |
| With husband /wife | 144 | 45.9 |
| With others | 29 | 9.2 |
| Source of income | | |
| No source of income | 12 | 3.8 |
| <1000 | 102 | 32.5 |
| 1001–2000 | 65 | 20.7 |
| 2001–3000 | 51 | 16.2 |
| 3001–5000 | 61 | 19.4 |
| >5000 | 23 | 7.3 |

**Table 2. Social support and substance use of the study participants at University of Gondar Hospital at Gondar town, Ethiopia, 2019 (n = 314).**

| Variables | | | Frequency | (%) |
|---|---|---|---|---|
| Ever used | Khat | Yes | 23 | 7.3 |
| | | No | 291 | 92.7 |
| | Alcohol | Yes | 57 | 18.2 |
| | | No | 257 | 81.8 |
| | Cigarette | Yes | 8 | 2.5 |
| | | No | 306 | 97.5 |
| Current used | Khat | Yes | 20 | 6.4 |
| | | No | 294 | 93.6 |
| | Alcohol | Yes | 54 | 17.2 |
| | | No | 260 | 82.8 |
| | Cigarette | Yes | 6 | 1.9 |
| | | No | 308 | 98.1 |
| Social support | | Poor | 115 | 36.6 |
| | | Moderate | 102 | 32.5 |
| | | Good | 97 | 30.8 |
| Past psychiatry history | | Yes | 5 | 1.6 |
| | | No | 309 | 98.4 |
| Past medical history | | Yes | 44 | 14 |
| | | No | 270 | 86 |
| Family history of mental disorder | | Yes | 17 | 5.4 |
| | | No | 297 | 94.6 |
| Medical OPD | | Yes | 265 | 84.4 |
| | | No | 49 | 15.6 |
| Orthopedic/surgical | | Yes | 49 | 15.6 |
| | | No | 269 | 84.4 |

sectional study while the lower report was a review of studies, the measurement tool we used was also not the same, moreover we only included the adult outpatient while the review study included study participants from different age group.

The finding of this study was also higher than the finding of a community-based survey in Ethiopia respondents based on ICD-10 criteria prevalence of (9.1%) [6]. The possible reason for this difference might be the use of different instruments and cutoff points to measure depression and study design. Conversely our, 15.9% was higher than the results of various studies, such as, 4.5% in Sri Lanka [4], 5.9% in Sri Lanka [5]. This difference might be because of the small sample size and population variations among the two study participants, differences in instruments may also be the case, as they employed BDI, HADS to identify the case while we used the patient health questionnaire-9. The other variation might be due to the methods they used for data collection; in Sri-Lanka large scale research and patient records were included. Our finding was higher than studies conducted in China(5.7%) [30], and Hong Kong(10.7%) [31]. This discrepancy might be due to the difference in the tool they used the Chinese version of Beck Depression Inventory (BDI), the study setup and socio-cultural variations among the study participants.

On the other hand, this finding was lower than different studies conducted in Ethiopia, like 29%Jima town residents [11]. It might also be due to socio cultural differences and tools used to measure depression with Beck Depression Inventory two (BDI-II).

**Table 3. Bivariate and multivariate logistic analyses results of study subjects (n = 314).**

| Variable | Depression | | | |
|---|---|---|---|---|
| | Yes | No | COR(95%CI) | AOR(95%CI) |
| **Occupation** | | | | |
| Farmer | 12 | 48 | 1 | 1 |
| House wife | 7 | 34 | 0.82(0.29–2.31) | 1.13(0.09–12.9) |
| Governmental worker | 10 | 64 | 0.63(0.25–1.57) | 1.19(0.10–14.04) |
| Private worker | 20 | 120 | 0.8(0.36–1.77) | 1.32(0.12–14.9) |
| Student | 1 | 18 | 0.22(0.03–1.83) | 1.39(0.14–13.3) |
| **Marital status** | | | | |
| Single | 17 | 121 | 1 | 1 |
| Married | 23 | 124 | 1.32(0.67–2.6) | 0.63(0.2–1.97) |
| Divorced | 10 | 19 | 3.75(1.49–9.3) | 1.55(0.17–14.4) |
| **Educational status** | | | | |
| Unable to Read& write | 15 | 33 | 1 | 1 |
| Able to read& write | 4 | 33 | 0.27(0.08–0.89) | 0.24(0.67–0.84) * |
| Primary Education (1–8) | 6 | 38 | 0.35(0.12–0.99) | 0.35(0.11–1.09) |
| Second Education (9–12) | 10 | 53 | 0.41(0.17–1.03) | 0.34(0.12–0.91) * |
| College/s University | 15 | 107 | 0.3(0.14–0.16) | 0.32(0.13–0.78) ** |
| **Living condition** | | | | |
| With another Person | 8 | 21 | 1 | 1 |
| Living alone | 16 | 56 | 0.75(0.28–2) | 1.03(0.33–3.11) |
| With parents | 4 | 65 | 0.16(0.44–0.59) | 0.32(0.79–1.29) |
| With Husband/wife | 22 | 122 | 0.47(0.19–1.2) | 0.74(0.26–2.1) |
| **Social support** | | | | |
| Strong | 5 | 12 | 1 | 1 |
| Moderate | 9 | 93 | 1.78(0.58–5.52) | 1.85(0.57–5.99) |
| Poor | 36 | 79 | 8.39(3.13–22.39) | 7.78(2.74–22.09) ** |
| **Substance use** | | | | |
| No current smoking | 46 | 262 | 1 | 1 |
| Current Cigarette smoking | 4 | 2 | 11.39(2.03–64.0) | 12.65(1.79–89.14) ** |

The result of this study was also lower than the finding inpatients admitted 24.5% in Hawassa [7] and38% south Ethiopia [32]. The possible reason might be the difference in the tool they used and setting variations.

Our finding was lower than pooled estimate prevalence of a systematic review and meta-analysis people living with HIV in Ethiopia both community and institution based study (36.65%) [33].

The findings of our study was lower than those of other institutions based cross sectional studies done in other countries53.2%inKochi [34], 23.8% in India [35], 30.3% in Malawi [36]. This variation may be due to the difference in study areas, clinical condition and socioeconomic status of participants, difference in the tool used.

According to the current study depression was significantly associated with educational status. Being able to read and write were 76%, secondary education were 66%, college/university education statuses were 68%times less likely for depression than illiterate. The result is in agreement with studies conducted in Ethiopia [37], South Africa [38], Sir Lanka [5]and Turkey [39]. The possible explanation for this could be the fact that individuals with illiterates were given less worth to their self-esteem and live a stressful life as compared with who have good educational status. In addition, educated people have better understanding of the risk factors

of depression compared to illiterates and even attending health service high among educated individuals.

Depression was significantly associated with social support. The odds of developing depression were 7.78 times higher among people with poor social support than strong social support. This result is consistent with different studies conducted in Ethiopia [10, 14, 17, 18, 40]. This might be due to the fact that poor social support may leads to social isolation, which can have a negative impact on mental and physical well-being.

Clients who had behavior of current cigarette smoker's were12.65 times more likely to develop depressive symptoms when compared to non-smokers. The finding was similar to the study conducted in Ethiopia substance users [11, 37]. Depression and smoking show bidirectional relationship, Substance use increases the risk of major depressive disorder [41], there are thousands of chemicals other than nicotine present in cigarette smoke, one or several may affect mood [42].

## Limitation of the study

A cross-sectional design cannot permit conclusions for some variables, for example, to decide whether the medical cases symptoms are risk for or a consequence for the undiagnosed depression.

## Conclusion

The current study showed that the prevalence of unrecognized depression among participants was high. Educational status, social support and current cigarette smoking were significantly associated with depression. Attention should be given in screening and treating depression, illiterate, poor social support and cigarette smokers. Further studies with longitudinal study design and other important variables should be considered. We highly suggest health care workers to screen patients for depression and training should be given to healthcare workers working in the medical outpatient department in order they recognize and manage depression accordingly or made referral.

## Supporting information

**S1 Fig.**
(DOCX)

## Acknowledgments

We acknowledge University of Gondar, College of Medicine and Health Sciences, department of Psychiatry for supporting the research on different ways. We are also thankful for the study participants and data collectors.

## Author Contributions

**Conceptualization:** Alemu Lemma, Haregewoyin Mulat, Wondale Getinet.

**Data curation:** Alemu Lemma, Kabtamu Nigussie.

**Formal analysis:** Alemu Lemma, Wondale Getinet.

**Investigation:** Alemu Lemma, Haregewoyin Mulat, Kabtamu Nigussie.

**Methodology:** Alemu Lemma, Haregewoyin Mulat, Kabtamu Nigussie.

**Resources:** Alemu Lemma, Haregewoyin Mulat.

**Software:** Wondale Getinet.

**Supervision:** Wondale Getinet.

**Validation:** Alemu Lemma, Haregewoyin Mulat.

**Visualization:** Wondale Getinet.

**Writing – original draft:** Alemu Lemma, Wondale Getinet.

**Writing – review & editing:** Alemu Lemma, Haregewoyin Mulat, Kabtamu Nigussie, Wondale Getinet.

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
