## [Decision Letter · Decision Letter 0]

26 Aug 2021

PONE-D-21-13384

Prevalence of Unrecognized Depression and Associated factors Among Medical Outpatient Department Attendees; a cross sectional study

PLOS ONE

Dear Dr. Lemma,

Thank you for submitting your manuscript to PLOS ONE. After careful consideration, we feel that it has merit but does not fully meet PLOS ONE’s publication criteria as it currently stands. Therefore, we invite you to submit a revised version of the manuscript that addresses the points raised during the review process.

ACADEMIC EDITOR: Please insert comments here and delete this placeholder text when finished. Be sure to:

Three reviewers have mixed recommendations.Please carefully address their comments.

We look forward to receiving your revised manuscript.

Kind regards,

Jianguo Wang, PhD

Academic Editor

PLOS ONE

Journal Requirements:

2. Thank you for including your ethics statement:  "Ethical clearance was obtained from the University of Gondar Institutional Review Board following the Ethiopian National Research Ethics Review Guideline of the Federal Ministry of higher education and Science. A permission letter was obtained from the Gondar referral hospital. Participants were fully informed about the purpose of the study ahead of the interview and informed consent was obtained from all participants.". 

Please provide additional details regarding participant consent. In the ethics statement in the Methods and online submission information, please ensure that you have specified what type you obtained (for instance, written or verbal, and if verbal, how it was documented and witnessed). If your study included minors, state whether you obtained consent from parents or guardians. If the need for consent was waived by the ethics committee, please include this information.

4. Thank you for submitting the above manuscript to PLOS ONE. During our internal evaluation of the manuscript, we found significant text overlap between your submission and the following previously published works, some of which you are an author.

-  https://bmjopen.bmj.com/content/bmjopen/9/6/e028550.full.pdf

-  https://bmcresnotes.biomedcentral.com/articles/10.1186/s13104-019-4673-6

-  https://preview-ijmhs.biomedcentral.com/articles/10.1186/1752-4458-6-23

-  https://www.hindawi.com/journals/drt/2016/3460462/?utm_source=google

The text that needs to be addressed involves the Discussion section.

Please revise the manuscript to rephrase the duplicated text, cite your sources, and provide details as to how the current manuscript advances on previous work. Please note that further consideration is dependent on the submission of a manuscript that addresses these concerns about the overlap in text with published work.

Reviewers' comments:

Reviewer's Responses to Questions

**Comments to the Author**

1. Is the manuscript technically sound, and do the data support the conclusions?

Reviewer #1: Yes

Reviewer #2: No

Reviewer #3: Yes

2. Has the statistical analysis been performed appropriately and rigorously? 

Reviewer #1: Yes

Reviewer #2: No

Reviewer #3: Yes

3. Have the authors made all data underlying the findings in their manuscript fully available?

Reviewer #1: Yes

Reviewer #2: No

Reviewer #3: Yes

4. Is the manuscript presented in an intelligible fashion and written in standard English?

Reviewer #1: Yes

Reviewer #2: No

Reviewer #3: Yes

5. Review Comments to the Author

Reviewer #1: 1.In the topic of your article you have mentioned the unrecognized Depression. Need to explain how unrecognized Depression is different from Depression.

2.In page no 3 third paragraph: the sentence is not clear and the Depression more common in females 4% than male 2.7% doesn't match with reference no 10.

3.Page no 3 paragraph 5 need to rewrite so that it will be more understandable.

4.In methods:

a. Duration of the study is not clear

b. inclusion and exclusion criteria not mentioned.

c. for sample size calculation why assumption prevalence of 24.5% has been taken even though you have mentioned that the prevalence of Depression in Ethiopia was 9% in National survey study?

d.PHQ-9 validated to use in local language reference not given.

e. Unstandardized semi structured questionnaires why translated back to English from Amharic? need further explanation.

f.why for bivariate analysis P value of less than or equal to 0.2 selected? need further explanation.

5.results:In bivariate and multivariate analysis why age of the patients was not included among other sociodemographic factors?

Reviewer #2: 1. Reviewer definitely recommends that the author should use an academic English editing service or ask for help from native English speakers with health research backgrounds. There are some grammar errors and many typos.

2. Please describe in detail how you perform systematic random sampling. What was the sample frame? How did you obtain the sample frame? What procedure did you perform to randomly select participants? This description is fundamental as it affects how you would interpret your results.

3. Why did you only collect data from May to June? Abstract said: May to June, but Methods said: March to June

4. The sample size of 314 seemed to be relatively small for this analysis and decreased to the precision (or efficiency) of estimates from the regression. 95% CI of current smoking status blew up to 89 because the study only had a few currently smoking participants.

5. You mentioned that the PHQ version you used had been validated; please cite those reliability and validation study

6. Please clarify the domains of the Oslo Social Support scale and how it was validated and/or modified in your study.

7. Regarding the prevalence of "unrecognized" depression, have you excluded people with a previous history of depression? Please clarify how you define “unrecognized”.

8. Using Stepwise method to select variables into the regression model is an antique statistical-driven method, which should not be used in this case to find associated factors. It would help if you tried using DAG (Directed acyclic graph) to guide your choices of variables to put into the model.

9. The author needs to discuss the strengths and limitations of the study.

10. Please refer to the STROBE checklist for correctly reporting a cross-sectional study: Microsoft Word - STROBE checklist cross-sectional.doc (equator-network.org)

Reviewer #3: General comment: This is a facility-based cross-sectional study that attempted to determine the prevalence of unrecognized depression and associated factors among medical outpatient attendees in a tertiary hospital in Ethiopia. The topic is interesting and of public health importance. However, more details are needed in some sections of the Methods for clarity, better understanding and reproducibility. The authors need to re-structure the discussion to enable good flow. I suggest that authors should edit the work as there are too many grammatical errors.

Additional information: Correct the N/A indicated under Ethics Statement as the work involves human participants and as such requires Ethics statement; which is already contained in the body of the work.

Abstract: Recommendation should be based on key findings. Revise. (P.2)

Background: Re-structure to bring out or add a note on justification for the study (p.4).

Methods:

Study area: Add a note on the outpatient clinics available (P.4)

Sampling technique: Give concise description of the systematic sampling technique which was used (P.4)

Data collection and analysis: How was prevalence of depression determined/defined? What component questions of patients’ Health Questionnaire (PHQ-9) were used to assess depression? How were they analyzed / graded to determine prevalence of depression? (P.5)

Results: Authors should add the table on assessment of prevalence of unrecognized depression (P.8)

The criteria for inclusion of variables into multivariable analysis contained under results should be moved to methods- data analysis (P.8)

Discussions: This should be re-organized to have good flow. What is the public health implication of the findings? What are the limitations of the study?

6. PLOS authors have the option to publish the peer review history of their article (what does this mean?). If published, this will include your full peer review and any attached files.

Reviewer #1: **Yes: **Dr. Dipak Kunwar

Reviewer #2: **Yes: **Linh Bui

Reviewer #3: No

---

## [Author Response · Author response to Decision Letter 0]

15 Oct 2021

September 10 /2021

Response to Editor and reviewers’ comments 

Manuscript ID PONE-D-21-13384

 Title: “Prevalence of Unrecognized Depression and Associated factors Among Medical Outpatient Department Attendees; a cross sectional study" 

Journal: PLOSE ONE 

Overall response 

First of all the authors would like to thank the editor/s and reviewers for taking time and giving us important feedback, comments and suggestions on our manuscript. Accordingly, we have considered all the feedback/comments as very important and we addressed them in the main manuscript document to improve the quality of the manuscript. Below, we present our point by point responses to all the comments, and suggestions provided by the reviewers. Kindly note that the changes in the revised version of the manuscript are highlighted using the track Changes made.

Editor Feedback

We are very grateful to the editor’s feedback to improve the quality of the manuscript. Kindly note that the changes in the revised version of the manuscript are highlighted using the track Changes made.

Authors’ response: Thank you very much for the suggestion, all style requirements are fulfilled as per PLOSE ONE standard template. 

2. …Please provide additional details regarding participant consent. In the ethics statement in the Methods and online submission information, please ensure that you have specified what type you obtained (for instance, written or verbal, and if verbal, how it was documented and witnessed). If your study included minors, state whether you obtained consent from parents or guardians. If the need for consent was waived by the ethics committee, please include this information

Authors’ response: 

Authors’ made correction as follows “Ethical clearance was obtained from the University of Gondar Institutional Review Board following the Ethiopian National Research Ethics Review Guideline of the Federal Ministry of higher education and Science. A permission letter was obtained from the Gondar referral hospital. The study was performed in accordance with the declaration of Helsinki. Participants were informed about the aim of the study, procedures of selection, and assurance of confidentiality, their names were not registered to minimize social desirability bias and enhance anonymity. The right to participate, to refuse or discontinue participation at any time they want and the chance to ask any thing about the study was given for the participants. Informed written consent was obtained from all participants.”

Authors’ response: We have removed from other section and included under Method section only.

4. Thank you for submitting the above manuscript to PLOS ONE. During our internal evaluation of the manuscript, we found significant text overlap between your submission and the following previously published works, some of which you are an author.

Authors’ response: We are very grateful for your suggestion. That happened because of we use the same reference, we have rephrased and made all necessary changes in the main document. 

Reviewer #1 

Kindly note that the changes in the revised version of the manuscript are highlighted using the track Changes made

1. In the topic of your article you have mentioned the unrecognized Depression. Need to explain how unrecognized Depression is different from Depression.

Authors’ response: We appreciated the reviewers’ curiosity. By unrecognizing depression we mean to say depression that didn’t identified or screened and got any treatment before. Authors’ wants to emphasis on the fact that the patients attend medical OPDs and get treatment exclusively for medical conditions and depression left unrecognized and untreated which we call it unrecognized depression. We considered depression as a disorder which is identified and put on treatment.

2. In page no 3 third paragraph: the sentence is not clear and the Depression more common in females 4% than male 2.7% doesn't match with reference no 10.

Authors’ response: we are grateful for the suggestion. We have made an important revision to the suggested paragraph including correction reference and we made the paragraph easy to understand as follows. “Depression is among the most common mental disorders characterized by sadness, loss of interest or pleasure, guilt feeling or low self-esteem, disturbed sleep or appetite, decreased energy, and poor concentration(6). Depression is common in life and can be in the form of blues or sadness, mourning, or in the form of hyperactivity and manic behavior(7). A one time, one-year and lifetime prevalence of depression among world population were reported to be 12.9%, 7.2% and 10.8% respectively(8). In Ethiopia prevalence of depression was reported to be 9%, and is the 7th leading cause of disease burden(6).”

3. Page no 3 paragraph 5 need to rewrite so that it will be more understandable.

Authors’ response:- We are great full for the suggestion we re-wrote it as follows “Comorbidity of depression with chronic medical conditions like diabetes, hypertension, asthma, sickle cell disease, cardiac diseases, chronic respiratory diseases and rheumatoid arthritis is very common (11). Studies reported the depression comorbidity with medical cased may result in prolonged hospital admission, increase physical symptoms, reduction in adherence to medical treatment and increased medical costs (7, 9).”

4. 

a) Duration of the study is not clear

Authors’ response: - We have clarified the study duration as “An institution based cross sectional study was employed from 22nd March to 30th April 2019.”

b) Inclusion and exclusion criteria not mentioned.

Authors’ response: - We have included, Inclusion and exclusion criteria as follows “All patients who attended adult medical OPD at University of Gondar compressive specialized hospital were the source population, and those who were attending adult medical OPD at University of Gondar compressive specialized hospital during the study period and who fulfilled inclusion criteria were considered as study population. Clients who have a prior diagnoses of depression, unable to communicate during the interview as a result of critical illness were excluded from the study” 

c) For sample size calculation why assumption prevalence of 24.5% has been taken even though you have mentioned that the prevalence of Depression in Ethiopia was 9% in National survey study?

Authors’ response: The 9% finding was national survey in which all ages’ groups and community members participated, with the focus of identifying prevalence of depression in the general population. In the current study, we conducted an institutional based study to determine prevalence of unrecognized or untreated depression among medical outpatient attendees, that’s why we took 24.5%, which was an institution based study conducted in aim of finding the prevalence of unrecognizing depression.

d) PHQ-9 validated to use in local language reference not given.

 Authors’ response: Thank you, we have included the reference in the main document. 

e) Unstandardized semi structured questionnaires why translated back to English from Amharic? Need further explanation. 

 Authors’ response: Thank you for asking us clarification question to make our paper more understandable. The questionnaires were translated back to English by another expert to insure its consistency with the original version and to know its understandability. We have indicated changes made in the original document.

f) Why for bivariate analysis P value of less than or equal to 0.2 selected? Need further explanation.

Authors’ response: We mainly used 0.2 to include as many variables as possible. It’s also a commonly used cuff off in our community the following studies are a few of the many studies done in our context by using P value less than or equal to 0.2 

https://ijmhs.biomedcentral.com/track/pdf/10.1186/s13033-019-0287-6.pdf

https://ijmhs.biomedcentral.com/track/pdf/10.1186/s13033-019-0274-y.pdf

https://www.ncbi.nlm.nih.gov/pmc/articles/PMC7125332/

https://www.hindawi.com/journals/drt/2021/1942674/

5. Results: In bivariate and multivariate analysis why age of the patients was not included among other sociodemographic factors?

Authors’ response: Thank you very much for the question, we have excluded the age as all our participants are adult and we have found no association on the study done on the same population from previous literatures, that’s why we only put the mean and SD of the participants age. 

Reviewer: # 2

Kindly note that the changes in the revised version of the manuscript are highlighted using the track Changes made

1. Reviewer definitely recommends that the author should use an academic English editing service or ask for help from native English speakers with health research backgrounds. There are some grammar errors and many typos.

Authors’ response: Thank you very much for the recommendation. The manuscript have been edited for language, punctuation, grammar etc by an academic expert in the field who have long experience of publishing academic articles. The changes were made throughout the manuscript main document.

2. Please describe in detail how you perform systematic random sampling. What was the sample frame? How did you obtain the sample frame? What procedure did you perform to randomly select participants? This description is fundamental as it affects how you would interpret your results.

Authors’ response: We are great full for the question. Our sample frame was a number of patients who visit medical outpatient department per month and we obtain the average number of patients visiting medical OPD per month and per year from Hospital administrative. Then we divided number of OPD visitors per month to our sample size to get the Kth value, we randomly choose the first study participant and then continued till we get a total of our sample size. We have included this in main document under Sample size and sampling procedure as follows “A systematic random sampling technique was used to select the study participants for interview. A total of 58,300 and 4,858 clients attend the medical OPD annually and monthly, respectively. The sampling fraction (K) was obtained by dividing monthly average number of patients attending medical out-patient department for the sample size, which is 15. The first individual was selected using a lottery method, and the rest were selected at a regular interval using systematic random sampling method.”

3. Why did you only collect data from May to June? Abstract said: May to June, but Methods said: March to June

Authors’ response: We are sorry for the inconsistency, we have corrected and made consistence data collection period both on the abstract and the method section.

We have collected the data from 22nd March to 30th April 2019. There was no specific reason why we choose the study period to be between March and April. 

4. The sample size of 314 seemed to be relatively small for this analysis and decreased to the precision (or efficiency) of estimates from the regression. 95% CI of current smoking status blew up to 89 because the study only had a few currently smoking participants

Authors’ response: Thank you for the question, dear reviewer the sample size of 314 seems small but we followed the scientific procedure to calculate the sample size for single population proportion formula and we added a non-response rate of 10%. 

5. You mentioned that the PHQ version you used had been validated; please cite those reliability and validation study.

Authors’ response: Thank you for the reminder, we have included the reference accordingly.

6. Please clarify the domains of the Oslo Social Support scale and how it was validated and/or modified in your study 

Authors’ response: Thank you for the question, however in the current study authors neither modified nor validated Oslo Social Support scale. We have added information about the tool in the main document as follow “… scale covers different fields of social support by measuring the number of people the respondent feels close to, the interest and concern shown by others, and the ease of obtaining practical help from others…” 

7. Regarding the prevalence of "unrecognized" depression, have you excluded people with a previous history of depression? Please clarify how you define “unrecognized”.

Authors’ response: We have excluded people who were on treatment for depression or who have a previous diagnosis of depression. By unrecognized depression, we mean outpatient medical attendees, who did not screened for depression before. 

8. Using Stepwise method to select variables into the regression model is an antique statistical-driven method, which should not be used in this case to find associated factors. It would help if you tried using DAG (Directed acyclic graph) to guide your choices of variables to put into the model.

Authors’ response: We are great full for the suggestion, unfortunately all authors are familiar with stepwise method that’s why we employed this method. But we acknowledge your recommendation. 

9. The author needs to discuss the strengths and limitations of the study. 

Authors’ response: Thank you for the suggestion, we have included the limitation and strengths of the study in the main document. 

10. Please refer to the STROBE checklist for correctly reporting a cross-sectional study: Microsoft Word

Authors’ response: We appreciate reviewers’ effort to make our paper better we have referred to the checklist and made correction as per the checklist. 

Reviewer #3

1. General comment: This is a facility-based cross-sectional study that attempted to determine the prevalence of unrecognized depression and associated factors among medical outpatient attendees in a tertiary hospital in Ethiopia. The topic is interesting and of public health importance.

Authors’ response: Thank you very much for your feedback.

2. The authors need to re-structure the discussion to enable good flow. I suggest that authors should edit the work as there are too many grammatical errors.

Authors’ response: We have re-structured the discussion part and the entire document has been edited for language, punctuation, grammar etc by an academic expert in the field who have long experience of publishing academic articles. The changes were made throughout the manuscript main document.

3. Correct the N/A indicated under Ethics Statement as the work involves human participants and as such requires Ethics statement; which is already contained in the body of the work. 

Authors’ response: We are really grateful for the recommendation and we have made correction. 

4. Abstract: Recommendation should be based on key findings. Revise. (P.2)

 Authors’ response: Thank you, we have made changes in the main document accordingly. 

5. Background: Re-structure to bring out or add a note on justification for the study (p.4).

Authors’ response: we have modified our justification as follows “…However, there are limited data in the country particularly unrecognized depression among medical patients where psychiatric services is limited and or not available is under studied. Therefore, this study aimed to assess the magnitude and associated factors of unrecognized depression among medical outpatient attendees. This would help for future integrated intervention and it would be an input of information for policymakers to think of intervention strategies.”

6. Study area: Add a note on the outpatient clinics available (P.4) 

 Authors’ response: We have added available OPDs in the main document under material and methods as “… the hospital has seven adult outpatient clinics and 600 inpatient beds…” 

7. Sampling technique: Give concise description of the systematic sampling technique which was used (P.4)

Authors’ response: - We have added description of how we employed systematic sampling technique in the main document as follows “A systematic random sampling technique was used to select the study participants for interview. A total of 58,300 and 4,858 clients attend the medical OPD annually and monthly, respectively. The sampling fraction (K) was obtained by dividing monthly average number of patients attending medical out¬patient department for the sample size, which is 15. The first individual was selected using a lottery method, and the rest were selected at a regular interval using systematic random sampling method.”

8. Data collection and analysis: How was prevalence of depression determined/defined? What component questions of patients’ Health Questionnaire (PHQ-9) were used to assess depression? How were they analyzed / graded to determine prevalence of depression? (P.5)

Authors’ response: We are grateful for the question, the authors determined the prevalence of depression using all items of PHQ-9. The tool contains 9 items each graded from 0 (not at all) to 3 (nearly every day). A cut of point of ten and above was used to determine the prevalence of depression. Those respondent who scored ten and above were identified as having depression. 

9. The criteria for inclusion of variables into multivariable analysis contained under results should be moved to methods- data analysis (P.8)

Authors’ response: - We have included inclusion and exclusion criteria as follows in the main document. “All patients who attended adult medical OPD at University of Gondar compressive specialized hospital were the source population, and those who were attending adult medical OPD at University of Gondar compressive specialized hospital during the study period and who fulfilled inclusion criteria were considered as study population. Clients who were already diagnosed with depression, unable to communicate during the interview as a result of critical illness were excluded from this study.”

10. Discussions: This should be re-organized to have good flow. What is the public health implication of the findings? What are the limitations of the study?

Authors’ response: We are grateful for the recommendation. We have re-organized our work and added limitation of the current study in the main document. 

With kind Regards,

Alemu Lemma, 

Corresponding author 

Email: emailmeonalex@gmail.com

---

## [Decision Letter · Decision Letter 1]

24 Nov 2021

Prevalence of Unrecognized Depression and Associated factors Among Medical Outpatient Department Attendees; a cross sectional study

PONE-D-21-13384R1

Dear Dr. Lemma,

We’re pleased to inform you that your manuscript has been judged scientifically suitable for publication and will be formally accepted for publication once it meets all outstanding technical requirements.

Kind regards,

Jianguo Wang, PhD

Academic Editor

PLOS ONE

Additional Editor Comments (optional):

Reviewers' comments:

Reviewer's Responses to Questions

**Comments to the Author**

1. If the authors have adequately addressed your comments raised in a previous round of review and you feel that this manuscript is now acceptable for publication, you may indicate that here to bypass the “Comments to the Author” section, enter your conflict of interest statement in the “Confidential to Editor” section, and submit your "Accept" recommendation.

Reviewer #1: All comments have been addressed

Reviewer #3: All comments have been addressed

2. Is the manuscript technically sound, and do the data support the conclusions?

Reviewer #1: Yes

Reviewer #3: Yes

3. Has the statistical analysis been performed appropriately and rigorously? 

Reviewer #1: Yes

Reviewer #3: Yes

4. Have the authors made all data underlying the findings in their manuscript fully available?

Reviewer #1: Yes

Reviewer #3: Yes

5. Is the manuscript presented in an intelligible fashion and written in standard English?

Reviewer #1: Yes

Reviewer #3: Yes

6. Review Comments to the Author

Reviewer #1: thank you very much for taking all the comments made by me positively and correcting your manuscript accordingly.

Reviewer #3: (No Response)

7. PLOS authors have the option to publish the peer review history of their article (what does this mean?). If published, this will include your full peer review and any attached files.

Reviewer #1: **Yes: **Dipak Kunwar

Reviewer #3: No

---

## [Editor Report · Acceptance letter]

10 Dec 2021

PONE-D-21-13384R1 

Prevalence of Unrecognized Depression and Associated factors Among Medical Outpatient Department Attendees; a cross sectional study 

Dear Dr. Lemma:

I'm pleased to inform you that your manuscript has been deemed suitable for publication in PLOS ONE. Congratulations! Your manuscript is now with our production department. 

Kind regards, 

on behalf of

Dr. Jianguo Wang 

Academic Editor

PLOS ONE